# Needs Assessment of Southeastern United States Vector Control Agencies: Capacity Improvement Is Greatly Needed to Prevent the Next Vector-Borne Disease Outbreak

**DOI:** 10.3390/tropicalmed7050073

**Published:** 2022-05-13

**Authors:** Kyndall C. Dye-Braumuller, Jennifer R. Gordon, Danielle Johnson, Josie Morrissey, Kaci McCoy, Rhoel R. Dinglasan, Melissa S. Nolan

**Affiliations:** 1Department of Epidemiology and Biostatistics, Arnold School of Public Health, University of South Carolina, 915 Greene Street #439, Columbia, SC 29208, USA; kyndallb@email.sc.edu (K.C.D.-B.); dmj2@email.sc.edu (D.J.); jam57@email.sc.edu (J.M.); 2CDC Southeastern Center of Excellence in Vector Borne Diseases, Gainesville, FL 32611, USA; kd.mccoy@ufl.edu (K.M.); rdinglasan@epi.ufl.edu (R.R.D.); 3Emerging Pathogens Institute, University of Florida, 2055 Mowry Rd, Gainesville, FL 32611, USA; 4Bug Lessons Consulting LLC, 1230 N. Pennsylvania St., Denver, CO 80203, USA; jennifer@buglessons.com; 5College of Veterinary Medicine, Department of Infectious Diseases & Immunology, University of Florida, Gainesville, FL 32608, USA; 6Emerging Pathogens Institute, Department of Infectious Diseases & Immunology, University of Florida, Gainesville, FL 32608, USA

**Keywords:** needs assessment-1, mosquito-2, tick-3, vector-borne disease-4

## Abstract

A national 2017 vector control capacity survey was conducted to assess the United States’ (U.S.’s) ability to prevent emerging vector-borne disease. Since that survey, the southeastern U.S. has experienced continued autochthonous exotic vector-borne disease transmission and establishment of invasive vector species. To understand the current gaps in control programs and establish a baseline to evaluate future vector control efforts for this vulnerable region, a focused needs assessment survey was conducted in early 2020. The southeastern U.S. region was targeted, as this region has a high probability of novel vector-borne disease introduction. Paper copies delivered in handwritten envelopes and electronic copies of the survey were delivered to 386 unique contacts, and 150 returned surveys were received, corresponding to a 39% response rate. Overall, the survey found vector control programs serving areas with over 100,000 residents and those affiliated with public health departments had more core capabilities compared to smaller programs and those not affiliated with public health departments. Furthermore, the majority of vector control programs in this region do not routinely monitor for pesticide resistance. Taken as a whole, these results suggest that the majority of the southeastern U.S. is vulnerable to vector-borne disease outbreaks. Results from this survey raise attention to the critical need of providing increased resources to bring all vector control programs to a competent level, ensuring that public health is protected from the threat of vector-borne disease.

## 1. Introduction

Arthropods affect human health by vectoring the causative agents of diseases such as malaria, dengue, lymphatic filariasis, and leishmaniasis. Worldwide, vector-borne diseases affect an estimated 320 to 620 million people annually, approximately 4 to 8 per cent of the world’s population [1]. The United States (U.S.) is not immune to the threat of vector-borne disease (VBD). In 2018, West Nile virus (WNV), Lyme disease (Lyme), and Zika virus (ZIKV) alone resulted in 36,387 confirmed human cases [2,3,4]. Of note, these diseases are only three of the eighteen nationally notifiable VBDs that threaten public health in the U.S. The southeastern U.S. is particularly vulnerable to VBD given that year-round high temperatures, humid environments, and increased outdoor recreation allow populations of vectors to thrive in areas with high human activity, resulting in increased risk of human–vector contact [5]. Further, poverty-related poor housing infrastructure can contribute to increased human–vector contact in the southeastern U.S., the national region with the highest poverty rates (>15.2% in nine states) [6,7]

Abiotic environmental factors ensure that the threat of vector-borne disease will likely increase in coming years. Global warming has increased the number of mosquito days (days when the temperature is between 50–95 °F and relative humidity is 42% or more) in many parts of the country [8], resulting in longer mosquito seasons and extended periods of time when people may encounter a potential vector. Additionally, the southeastern U.S. is vulnerable to new and established invasive mosquitoes such as *Aedes albopictus* Skuse, *Ae. aegypti* L., *Ae. japonicus* Theobald, and *Ae. scapularis* Rondani, competent vectors of several arboviruses of public health importance including ZIKV, dengue virus, and La Crosse encephalitis virus [5,9,10,11,12,13,14]. In 2018, the threat from *Ae. aegypti* and *Ae. albopictus* was realized in the southeastern U.S. when that region accounted for 27.0% of all U.S. ZIKV infections [4]. Further, the southeastern U.S. has recently noted increased abundance of several tick species [15,16,17,18]. This threat can be compounded when in some instances a single tick species can vector the causative agents of several diseases. For example, the blacklegged tick, *Ixodes scapularis Say,* has an ever-expanding range and is capable of vectoring the causative agents of Lyme disease, anaplasmosis, babesiosis, *Borrelia miyamotoi* disease, Powassan virus disease, and ehrlichiosis [15].

One critical tool in interfering with vector-borne disease transmission is eliminating populations of vectors through control efforts by municipal and private organizations. Yet, successful vector control programs are complex and must be properly resourced to provide protection to the public. 

One national study found that 84% of mosquito control agencies needed improvement in at least one area of core competency [19]. Furthermore, the unmet needs of vector control programs can be exacerbated by emerging public health threats and natural disasters that divert resources away from vector control to respond to more immediate threats. For example, hurricanes and pandemics can divert resources from vector control and public health agencies to other needed areas, resulting in reduced staff available to perform surveillance and control services [20,21]. Therefore, regularly assessing organizational capacity and infrastructure need is a critical step in ensuring that vector control programs have sufficient resources by serving as a justification to obtain funding and resources critical to addressing gaps in current efforts. The purpose of this study was to assess the needs of vector control in the southeastern U.S. and provide a justification for funding vector control and research support. Of note, the execution of the survey coincided with the inception of the COVID-19 pandemic. Subsequently, the needs assessment also provided insights into how a competing public health threat impacts vector control. 

## 2. Materials and Methods

### 2.1. Survey Development

The needs assessment survey was developed based on the framework provided by the WHO [22]. Based on those recommendations, the objectives of the survey were to assess: the current status of VBD in 13 states of the southeastern U.S. (South Carolina, North Carolina, Tennessee, Georgia, Florida, Alabama, Mississippi, West Virginia, Virginia, Kentucky, Louisiana, Arkansas, and Missouri), the risk of future VBD outbreaks, vector species, risk of introducing invasive vectors, current capacity to respond to future vector-borne outbreaks, status and capabilities of surveillance systems, organizational need, and opportunities for program improvements.

The survey was reviewed by the University of South Carolina’s Institutional Review Board and was determined as exempt of human subject’s research given the anonymous survey respondent design. The survey targeted public health practitioners, vector control districts and associations, integrated pest management researchers and educators, and state emergency preparedness staff. A one-page (front and back) questionnaire (Appendix A) was created in collaboration between entomologists and epidemiologists experienced in applied vector control. The survey consisted of 33 multiple choice, Likert scale, and free text-response questions and was designed to measure the risk factors, perceived and actual needs, practices, behaviors, and possible improvements that could be made within the communities regarding mosquito and tick control and spread of VBD. 

### 2.2. Identification of Recipients

The e-mail and physical mailing addresses of vector control directors and personnel were compiled into a database by performing a thorough Google (Alphabet Inc., Menlo Park, CA, USA) search of state and regional mosquito control associations and using their member databases. Additional recipients were found by compiling the county information for the 13 southeastern states and retrieving vector control individuals from the respective county websites. Finally, city government websites were investigated, and individuals responsible for vector control identified. Recipient information was compiled into a database of 386 unique contacts. Recipients included: government agencies (state, city, and county health departments, divisions/bureaus of environmental health, public works, and waste management), local vector control associations, and regional academic institutions.

### 2.3. Distribution

A paper and electronic version of the same survey was created. Paper copies were sent through the U.S. Postal Service, and envelopes were hand addressed to optimize response. A cover letter informing the participants of the objectives of the project and a pre-paid postage envelope with a completed return address were included with the survey. An electronic version of the survey was created using the online software program Google Forms. 

An e-mail invitation to participate in the survey was sent on 26 February 2020 with the subject heading: ‘University of South Carolina and CDC Needs Assessment for Emerging Vector-Borne Disease Threats in the Southeastern United States’. A follow-up reminder was sent on 26 March 2020. A physical copy of the questionnaire was mailed to non-respondent contacts on 3 April 2020. Newly completed surveys were no longer accepted starting 1 June 2020. In total, 383 contacts received an email with a link to the survey, and 384 contacts received a physical copy of the survey. Surveys were collected from March to May 2020. All responses remained anonymous. 

### 2.4. Statistical Analysis

Data were cleaned and converted to binary variables. Initial Likert scale or ordinal variables were individually assessed for univariate statistical relevance with dependent variables and were all eventually classified as binary variables for streamlined multivariate analysis. Univariate logistic regressions were performed. Multivariate backwards stepwise regression models were under-powered and therefore not presented. All statistical methods were employed in STATA v.15 (College Station, TX, USA). 

## 3. Results

### 3.1. Survey Responses and Respondent Characteristics

A 39% response rate (150 completed surveys) was received with respondents from 12 out of the 13 states assessed. Florida, North Carolina, and Virginia had the most respondents ranging from 15–36 per state. Kentucky, South Carolina, and West Virginia had 11–14 respondents per state. Alabama, Louisiana, and Missouri had 4–10 respondents per states. Arkansas, Georgia, Mississippi, and Tennessee had the lowest number of respondents with 3 per state. 

Among the returned questionnaires, 78 responses came from Google Forms, and 72 came from returned paper forms. Respondents’ highest educational training levels were high school (28%), bachelor’s degree (50%), or graduate degree (22%). Less than half of respondents worked for agencies that were part of a local health department (47%). The majority of respondents worked at the county level (70%), with the remainder of the respondents working at city (18%), state (9%), and regional (3%) levels. Approximately half of the respondents worked for agencies that serviced a population area greater than 100,000 residents (48%).

### 3.2. Vector Surveillance Results

The most commonly reported mosquito species among all participating agencies were *Ae. albopictus* (91%) and *Anopheles quadrimaculatus* Say (70%). Other mosquito species reported include:* Culiseta melanura* Coq. (68%), *Culex pipiens* L. (61%), *Ae. triseriatus* Say (53%), *Cx. restuans* (51%), *Ae. aegypti* (45%), and *Cx. nigripalpus* Theobald (42%; Figure 1A). The most commonly reported tick species among all participating agencies were *Ixodes scapularis* (57%), *Amblyomma americanum* L. (56%) and *Dermacentor variabilis* Say (54%). Other tick species included: *Am. maculatum* Koch, *Rhipicephalus sanguineus* Latreille, and *Haemaphysalis longicornis* Neumann (Figure 1B). 

The number of different mosquito and tick species reported varied by state. In general, more mosquito species relative to tick species were reported in the southernmost states; however, fewer mosquito species relative to tick species were reported in the more northern states (Figure 2). Of note, *Aedes* spp. was reported in all 13 states. 

### 3.3. Vector-Borne Diseases Reported

The most reported mosquito-borne virus detected was West Nile virus (85%) followed by Eastern equine encephalitis virus (56%), St. Louis encephalitis virus (27%), and La Crosse encephalitis virus (13%). The most commonly reported tick-borne disease was Lyme disease (44%) followed by spotted fever group rickettsioses (11%). Additional VBD reported were Chagas disease (4%), murine typhus (1%), and unknown others (16%).

### 3.4. Agency Demographic Associated Capabilities

The two primary dependent variables were (1) agency size of above or below 100,000 residents (Table 1) and (2) agency part of a local health department or not affiliated with a health department (e.g., public works, environmental services, etc. (Table 2). Agencies servicing larger communities were 3 times more likely (*p* = 0.005) to perform pathogen testing and 4 times more likely (*p* < 0.001) to perform year-round surveillance. Interestingly, a large resident number was negatively associated with tick surveillance, yet agencies servicing larger communities were significantly more likely to perform in-house geographic information system (GIS) mapping (*p* < 0.001; Figure 3), own major equipment (*p* = 0.010), conduct vector species identification (*p* < 0.001), and perform in-house insecticide resistance testing (*p <* 0.001). Agencies affiliated with health departments were 5 times more likely (*p* = 0.034) to perform tick surveillance. Additionally, public health-affiliated organizations were more likely to own insecticide application trucks (*p* = 0.002) and less likely to conduct GIS and mapping in-house (*p* < 0.027). 

## 4. Discussion

The combination of progressive environmental shifting to subtropical climates, expansion of invasive vectors, diffuse poverty, and increased international travel make the southeastern U.S. particularly vulnerable to vector-borne disease from mosquitoes and ticks [23]. In response to these threats, many areas establish vector control programs to interfere with disease transmission by eliminating vectors. The CDC and American Mosquito Control Association describe five core capabilities of competent programs: routine surveillance, evidence-based treatment decisions, adulticiding/larviciding, routine utilization of several methods to control vectors, and insecticide-resistance monitoring [24]. The current needs assessment found that agencies serving greater than 100,000 residents and those associated with a public health department performed more core and supplemental competencies (Figure 3) compared to smaller agencies or agencies not affiliated with a public health department. Given that more than half of respondents were smaller (52%) or not affiliated with a public health department (53%), these findings suggest that the majority of the southeastern U.S. is vulnerable to vector-borne disease threats. 

This assessment identified inconsistencies among responding vector control agencies in relation to controlling for vectors, especially mosquitoes. Once the threshold to justify the use of an adulticide has been reached, vector control programs currently have the choice between two classes of insecticides: pyrethroids and organophosphates [24]. Pyrethroids are the most commonly used class of insecticides in the world [24,25], so the high reported use of permethrin, a pyrethroid, found in this survey is unsurprising. More interesting is the association between large agencies and the utilization of several active ingredients and formulas (malathion, an organophosphate, and larvicides) targeting both adult and immature mosquitoes, which could suggest that larger agencies have the resources to utilize more tools at their disposable. However, the most important finding related to the chemical control of vectors is that less than half of all respondents perform insecticide resistance monitoring. The current finding, again, suggests that the majority of vector control programs in the southeastern U.S. are not competent and subsequently vulnerable to VBD threats. 

Given the paucity of active ingredients currently available for adulticide and the poor prospects of a new class of insecticide being registered for this purpose [26], insecticide-resistance monitoring is critical to keep the currently available active ingredients viable. A previous national needs assessment found similar results regarding a lack of insecticide resistance monitoring [19], implying that this is not just a regional problem, but a national one as well. These implications encourage action to rectify the situation immediately by providing more resources from the federal, state, and local levels. International models exist that serve as examples of how to leverage minimal resources to establish an effective insecticide resistance monitoring program. For example, American vector control agencies would benefit from discussions with and implementation of capacity and infrastructure programs from Mexico’s Health Secretary [27,28]. 

In September 2020, the CDC released *A National Public Health Framework for the Prevention and Control of Vector-Borne Diseases in Humans* and highlighted that seven out of the nine newly reported vector-borne diseases in the U.S. since 2004 were tick-borne [29]. Thus, the CDC effectively promoted that vector control programs should incorporate tick surveillance into their regular activities. Potential consequences of low tick surveillance include the unidentified introduction of invasive species, emerging pathogens, and underdiagnoses of known pathogens [30]. The current needs assessment is the first regional assessment to include specific questions about ticks and revealed a troublingly low number of tick surveillance programs amongst all respondents (8%). This is in line with the results from a national survey conducted in 2019 to assess tick surveillance capabilities where the southeast U.S. had the lowest proportion of programs conducting active routine tick surveillance and financial support for pathogen testing in ticks [31]. Interestingly, this national survey revealed that the southeastern region was the most concerned about the introduction of invasive tick species. This is relevant as the southeastern states are the southernmost edge for reported Asian longhorned tick (*H. longicornis*) spread nationally [32].

We found a negative relationship between tick surveillance and agency size, suggesting that agencies serving larger communities were less likely to perform tick surveillance. More work needs to be performed to understand this trend, but the negative association may be due to funds provided to larger associations designated for specific activities, thus preventing resources from being allocated to tick control efforts. Alternatively, larger districts may be associated with urban centers and feel the risk of tick-borne threats does not warrant the expenditure of funds. Finally, respondents affiliated with a local health department were 5.5 times more likely to perform tick surveillance and control, possibly suggesting that departments with a broader mission have more flexibility about developing vector control programs that target both mosquitoes and ticks. 

Resources dedicated to addressing VBD have a history of being shunted during times of competing disasters [20]; however, some natural disasters may create favorable conditions for increased abundance of vectors [33,34], thus generating cyclical and often regional disparities in funded vector-control efforts. Funding increases post-Hurricanes Matthew, Irma and Harvey serve as examples [20,35,36]. The current needs assessment survey corresponded with the COVID-19 pandemic and shed light on vector control needs during a competing emerging public health crisis. While similarities between the findings in this survey and the 2017 National Association of County and City Health Officials (NACCHO) report suggest that some identified gaps existed before the pandemic [19], some gaps were certainly exacerbated over the past year due to a finite amount of resources being allocated between competing interests (vector-borne disease control versus COVID-19 response). For example, the CDC Division of Vector-Borne Disease has experienced a diversion of resources as numerous members of staff, including the division’s director, assisted with the CDC’s response to COVID-19 [37]. Thus, work should be carried out to establish a new baseline after the pandemic to serve as a benchmark to evaluate the effectiveness of future vector control efforts. Additionally, the pre- and post-pandemic baselines should be compared to understand how competing public health threats impact vector control. The results from comparing these baselines could be used to justify resources during future natural disasters to mitigate the risk of simultaneous vector-borne disease outbreak.

The best vector control programs utilize an integrated approach beginning with the surveillance of vectors and pathogens [38]. In the southeastern U.S. where environmental conditions allow populations of mosquitoes and ticks to occur all year in some areas, vector surveillance programs need to occur year-round as well. However, less than half of respondents performed year-round surveillance (Table 1). The consequences of seasonal surveillance are great and could include higher risk of VBD outbreaks and delayed response when outbreaks do occur due to delayed detection of vectors and/or pathogens. Additionally, seasonal vector control programs exacerbate the need for qualified personnel since seasonal work results in high turnover and a lack of trained, educated workforce. Allocating additional funding to surveillance could allow smaller vector control programs to expand their capabilities or institutions such as the CDC Centers of Excellence in Vector Borne Diseases to develop innovative technologies, such as remote sensing, Wolbachia-infected *Aedes* spp. release or establishing local genetically modified mosquito populations, that require less personnel to find and identify a vector species [39,40,41]. Further, international colleagues in Brazil, Mexico, Vietnam, South Sudan, and other countries highlight novel lessons learned for increasing effective integrated vector management programs in rural, impoverished areas [28,42,43,44].

Whenever possible, surveillance programs should include a pathogen detection component; however, pathogen testing is expensive and can require highly specialized equipment or funds dedicated to paying third party laboratories to perform screenings. Screening for pathogens was low among all respondents of this survey, and even though pathogen testing was associated with programs serving larger areas, less than half of these larger agencies were performing such testing. Collaborations with local academic partners or new dedicated government funding lines have the potential to boost vector pathogen surveillance. The consequences of not performing pathogen testing include undetected transmission, missed pathogen introduction, and improper use of insecticides. Additionally, action thresholds for treatment will be different based on the type of testing. For instance, during the Hurricane Harvey disaster response, 30 mosquitoes landing on an individual in 1 min, resulting in control measures [20]; however, the risk presented by just a single pool of mosquitoes testing positive for a pathogen represented justification to apply adulticides [20]. Thus, sufficiently resourcing vector control programs so they can include pathogen surveillance data is critical to protecting public health and proper resource allocation. 

Vector control programs take surveillance data to create maps that visualize information to help make evidence-based decisions and inform the public regarding risk and treatment applications. To maximize this process, real-time feedback tools that identify vector geospatial hotspots are critically needed. Only 64% of respondents performed in-house mapping and GIS, and this capability was biased toward larger programs. With most agencies having a GIS trained person, the opportunity to develop rigorous prediction-based models using the discipline standard software exists. A CME-based half-day training has promise for the integration of evidence-based protocols that could predict hotspots for insecticide targeting. Having preexisting maps with these demographic areas already identified and trained experts onsite could allow a vector control program to respond more rapidly to an evolving threat. Developing remote sensing technology that can upload data and generate neighborhood-level precision prediction maps could automatically allow for faster response times and overall lower facility costs. Alternatively, vector control programs could investigate partnerships with their local city governments to utilize ArcGIS capabilities that may be available to the city. We hypothesize that the finding that vector control agencies associated with a health department are significantly less likely to have a GIS trained employee is due to the high ArcGIS uptake among their counterparts: parks department, public works, city planning, etc.

The current needs assessment survey is not without limitations. The majority of respondents reported detection of WNV, but this could be due to the historical impact of WNV. Another limitation could be the 39% response rate, which may be due to the onset of the COVID-19 pandemic. Surveys were mailed in March 2020, a few days before many organizations faced government-mandated closure. Consequently, survey responses may have been hindered, and therefore the study naturally lacked some participation. Thankfully, this problem was mitigated by sending out an email version of the survey. Another possible limitation was that the survey was constructed for a targeted audience. Results should not be interpreted to reflect the viewpoints or opinions of other sub-sample groups, including individuals working within the medical community. Additionally, a small number of comments provided by the respondents indicated that the survey was not always given to the most appropriate person at each agency. Finally, the survey witnessed attrition of responses across the length of the questionnaire. No question items were flagged as mandatory to respond, so of the 150 respondents included in the sample, some respondents did not fully complete the question series, resulting in the denominator for each individual question to vary.

The only way to protect Americans from the threat of VBD is to provide enough resources so all vector control programs perform at the competent level. Organizations must continue assessment of the need to guide policy makers in establishing or strengthening their capacity and capability for vector control in a way that coordinates multiple sectors and leverages data for local adaptability. Taken as a whole, the results from this and the national survey show that more funding must be allocated to all vector control programs, particularly to the agencies not affiliated with health departments and servicing smaller populations. Recent legislation authorizing funding for mosquito (Pandemic and All-Hazards Preparedness and Advancing Innovation Act of 2019) and tick control (Kay Hagan Tick Act of 2019) are steps in the right direction; however, at the time of writing this publication, neither bill was fully appropriated, resulting in an inadequate level of funding. During times of respite from vector-borne disease, the cost of vector control can seem unnecessary compared to competing interests; however, this upfront investment not only increases public health, but it could also save money by preventing heavy economic impacts from emerging and endemic diseases. One model investigating the potential economic impact of ZIKV found that just a 0.01% attack rate in six states of the U.S. would cost an estimated USD 184.4 million dollars to society through direct medical costs and loss of productivity [45]. Similarly, Lyme in the U.S. has been estimated to cost up to USD 786 million per year [46]. Thus, the return on investment of a vector control program through cost savings and health benefits to the public further justifies addressing the gaps identified in the current needs assessment and providing increased resources to vector control programs.

## 5. Conclusions

A March 2020 needs assessment implemented in the southeastern U.S. revealed that most local vector control agencies failed to meet the basic core requirements for effective insect management. Smaller townships (<100,000 people) and agencies not affiliated with a public health department were particularly vulnerable to failing capacity and infrastructure. Despite ticks being a leading cause of vector-borne disease in the U.S., 8% of southeastern U.S. vector control agencies surveil and control for this species. Specifically, most agencies reported not performing pathogen or insecticide resistance testing in-house, yielding potential for undetected outbreaks. To address these infrastructural gaps, funding is warranted to support programs such as local vector control and regional institutions with the mission to protect the public from vector-borne disease threats. 

## Figures and Tables

**Figure 1 tropicalmed-07-00073-f001:**
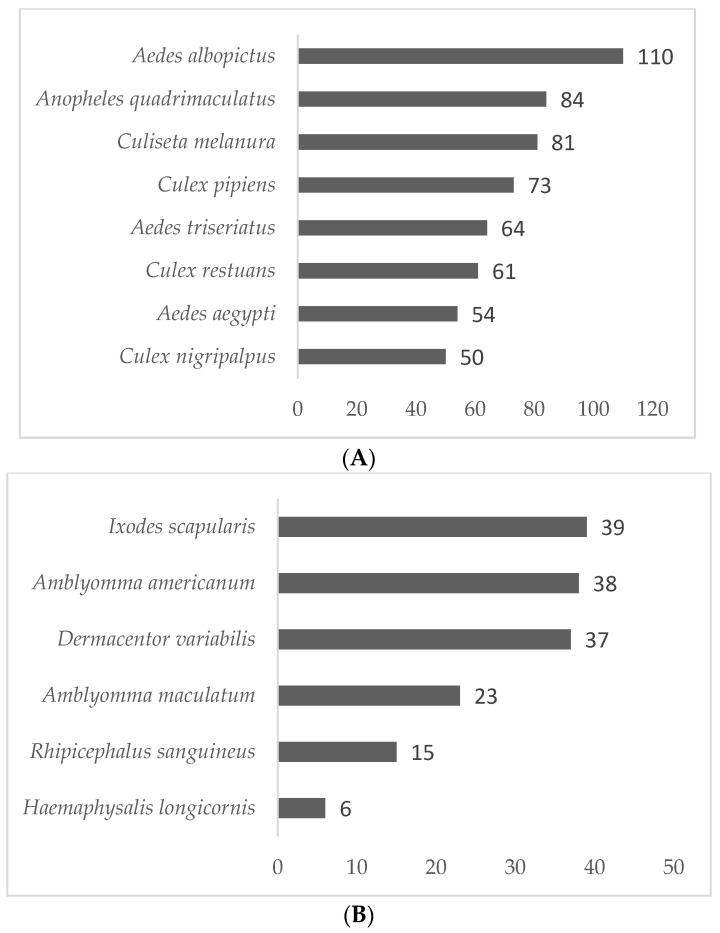
Reported mosquito and tick species found during surveillance efforts (*n* = 150 agencies). (**A**) Number of organizations reporting mosquito species in their region. (**B**) Number of organizations reporting tick species in their region.

**Figure 2 tropicalmed-07-00073-f002:**
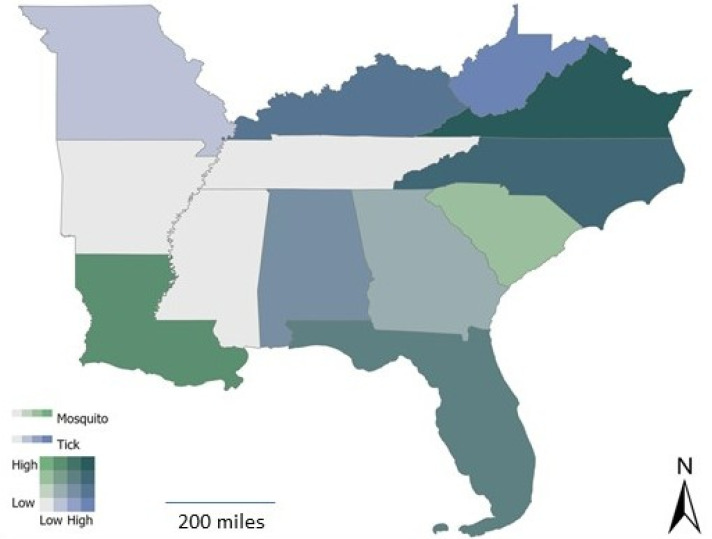
Reported number of mosquito species versus number of tick species reported in each state.

**Figure 3 tropicalmed-07-00073-f003:**
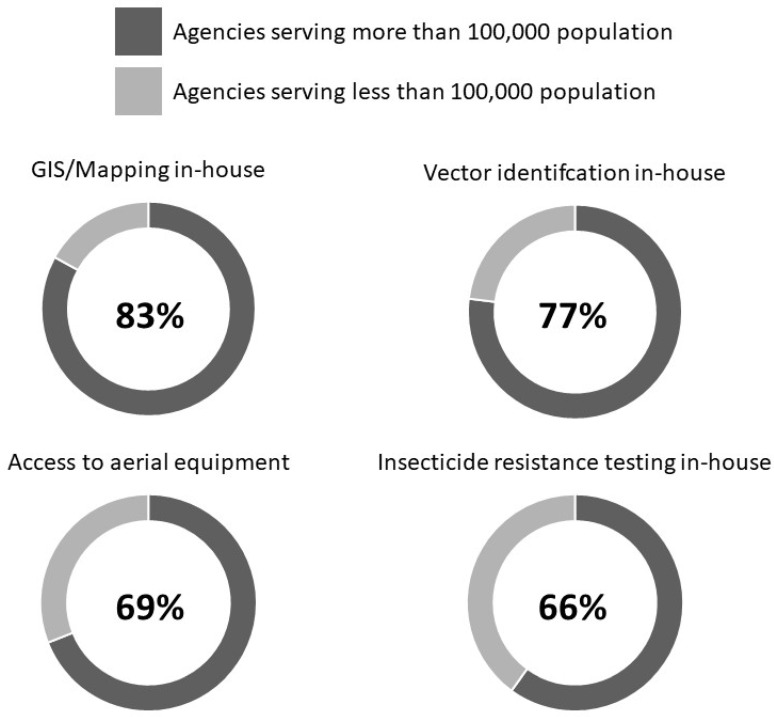
Agencies serving larger communities (>100,000) have significantly more access to resources.

**Table 1 tropicalmed-07-00073-t001:** Organizational capacity stratified by service area population size.

	All Participants	Agency Size > 100,000 Residents ^1^	
	Number (%)	Number (%)	*p*-Value ^2^; OR ^3^ (95% CI)
**Vector(s) agency controls for:**			
Mosquitoes	136 (96%)	69 (99%)	0.136
Ticks	11 (8%)	2 (3%)	0.046; 0.20 (0.04–0.97)
Other ^4^	20 (14%)	9 (13%)	0.654
**Surveillance conducted all-year** (vs. summer only)	63 (49%)	41 (65%)	<0.001; 3.82 (1.83–7.96)
**Surveillance type:**			
Vector collections	92 (70%)	46 (71%)	0.848
Pathogen testing	39 (30%)	27 (42%)	0.005; 3.14 (1.41–6.97)
**Adulticides used:**			
Malathion	24 (20%)	17 (27%)	0.044; 2.70 (1.03–7.10)
Permethrin	99 (83%)	49 (79%)	0.324
**Larvicides used:**			
Biological control	81 (66%)	52 (81%)	<0.001; 4.49 (1.99–10.14)
Growth regulators	76 (62%)	48 (75%)	0.002; 3.33 (1.55–7.19)
Contact insecticides	41 (34%)	27 (42%)	0.043; 2.24 (1.03–4.89)
Stomach insecticides	61 (50%)	37 (58%)	0.056
**Insecticide applied at least biweekly**	45 (35%)	28 (41%)	0.031; 2.21 (1.08–4.53)
**Major equipment:**			
Organization-owned truck	106 (81%)	58 (88%)	0.057
Organization-owned aerial	35 (27%)	24 (36%)	0.010; 3.03 (1.31–7.03)
Contractor	19 (15%)	11 (17%)	0.526
**Conducts vector speciation in-house**	81 (56%)	54 (77%)	<0.001; 6.23 (2.99–12.98)
**Conducts disease testing in-house**	9 (6%)	5 (7%)	0.683
**Performs community outreach and education**	120 (83%)	61 (87%)	0.169
**Conducts GIS or mapping in-house**	94 (64%)	59 (83%)	<0.001; 5.78 (2.68–12.50)
**Conducts insecticide resistance testing in-house**	54 (44%)	37 (66%)	<0.001; 5.50 (2.51–12.02)

^1^ Dependent variable representing the size of an area serviced by a vector control program. ^2^ Univariate logistic regressions were performed, and all statistical methods were employed in STATA v.15 (College Station, TX, USA). ^3^ OR stands for odds ratio. ^4^ Other pests controlled included kissing bugs, bed bugs, rats, and sand flies.

**Table 2 tropicalmed-07-00073-t002:** Organizational capacity stratified by agency type.

	All Participants	Agency Part of Local Health Department ^1^	
	Number (%)	Number (%)	p-Value ^2^; OR (95% CI) ^3^
**Vector(s) agency controls for:**			
Mosquitos	136 (96%)	62 (46%)	N/A
Ticks	11 (8%)	68 (48%)	0.034; 5.49 (1.14–26.41)
Other ^4^	20 (14%)	12 (60%)	0.246
**Surveillance conducted all-year** (vs. summer only)	63 (49%)	29 (46%)	0.989
**Surveillance type:**			
Vector collections	92 (70%)	40 (43%)	0.278
Pathogen testing	39 (30%)	16 (41%)	0.409
**Adulticides used:**			
Malathion	24 (20%)	6 (25%)	0.099
Permethrin	99 (83%)	39 (39%)	0.769
**Larvicides used:**			
Biological control	81 (66%)	34 (42%)	0.754
Growth regulators	76 (62%)	31 (41%)	0.955
Contact insecticides	41 (34%)	18 (44%)	0.641
Stomach insecticides	61 (50%)	19 (31%)	0.028; 0.44 (0.21–1.71)
**Insecticide applied at least biweekly**	45 (35%)	17 (38%)	0.127
**Major equipment:**			
Organization-owned truck	106 (81%)	39 (37%)	0.002; 0.23 (0.09–0.60)
Organization-owned aerial	35 (27%)	13 (37%)	0.376
Contractor	19 (15%)	9 (47%)	0.714
**Conducts vector speciation in-house**	81 (56%)	34 (42%)	0.089
**Conducts disease testing in-house**	9 (6%)	5 (56%)	0.607
**Performs community outreach and education**	120 (83%)	59 (49%)	0.405
**Conducts GIS or mapping in-house**	94 (64%)	38 (40%)	0.027; 0.56 (0.23–0.92)
**Conducts insecticide resistance testing in-house**	54 (44%)	23 (43%)	0.126

^1^ Dependent variable representing a vector control program’s association with a public health department or not. ^2^ Univariate logistic regressions were performed, and all statistical methods were employed in STATA v.15 (College Station, TX, USA). ^3^ OR stands for odds ratio. ^4^ Other pests controlled included kissing bugs, bed bugs, rats, and sand flies.

## Data Availability

The generated survey dataset is available upon written request to the corresponding author.

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
