# Peer review of "Needs Assessment of Southeastern United States Vector Control Agencies: Capacity Improvement Is Greatly Needed to Prevent the Next Vector-Borne Disease Outbreak"

_tropicalmed, 2022, doi:10.3390/tropicalmed7050073_

Round 1

Reviewer 1 Report

This study is relevant to be considered by authorities to unify criteria and methods of Southeastern United States Vector Control Agencies.

Introduction. Include more citations in the last three paragraphs.

Lines 42-45. Include indoor factors (e.g., house design) along with environmental conditions.

Line 159. Describe the acronym of USPS.

Line 196. Ae. albopictus italicized.

Line 203. Fig. 1 A-B. I suggest including which mosquito and tick species are present by state.

Line 276. Please Include preventive before proactive.

Line 288. Please include in the discussion how to improve insecticide resistance monitoring, as well as other strong programs, such as the Mexican one.

 chrome-extension://efaidnbmnnnibpcajpcglclefindmkaj/https://www.gob.mx/cms/uploads/attachment/file/598088/Manual_de_Organizaci_n_y_Procedimientos_de_las_UIEB_s.pdf)

chrome-extension://efaidnbmnnnibpcajpcglclefindmkaj/https://www.gob.mx/cms/uploads/attachment/file/598093/Guia_para_la_Determinaci_n_de_la_SusceptibilidadResistencia_y_Eficacia_..._compr.pdf)   

Lines 331-332. I suggest mentioning the examples of Hurricane Matthew and Irma (2016 and 2017 respectively) (Weaver JR, Xue R, Gaines MK. Population outbreaks of mosquitoes after hurricanes Matthew and Irma and the control efforts in St. Johns County, Northeastern Florida. J Am Mosq Control Assoc. 2020;36:28–34. doi: 10.2987/19-6867.1.).

Lines 348-356. Please include some successful IVM programs elsewhere that could be implemented in the US.

Lines 359-360. Include examples of innovative strategies like Wolbachia-infected or/and GM mosquitoes in Florida.

Lines 372-374. Discuss the possibility of proposing collaboration with universities or the creation of state or federal labs for pathogen detection.

Lines 375-395. Discuss the use of hot-spots for diseases to maximize resources.

Lines 403-405. I agree and justify the need to create a national or regional plan to unify all criteria (not based on points of view) used for vector control programs.

Lines 413-418. Authors should include the mechanisms to present a solid or robust IVM program, and an adequate budget should be allocated to ensure the proper functioning and operating of the program.

Discuss marginalization and the generation of socio-economic status and public health risks in the Southeastern United States.

Supplementary

The effectiveness of vector control continuing education programs should be considered.

Reviewer 2 Report

The article has significance. It is interesting, however, the discussion is too long and the reader will lose interest.

1-It was a need assessment survey for the capacity improvement to prevent the next vector-borne diseases outbreaks in the Southeastern U.S. region. To prevent vector-borne diseases, zoonotic diseases, and even epidemics and pandemics, need assessment surveys play a crucial role to develop and implement the strategic framework to curb the diseases and pandemics. Therefore, I categorized it as interesting and important. 
2-I am not sure about any other study published in the Southeastern U.S. region (that has a probability of novel vector-borne disease introduction according to authors). However, need-based surveys are important for gap analysis for the success of any vector control and vector-borne disease control program.
3.-The topic is important however the paper is too lengthy, the text in the introduction gave me the impression sometimes that it is part of discussion and recommendations. The same case is about the discussion. It is too long. Authors can make it concise and then give recommendations and conclude findings. 

4- The authors did not write conclusions separately, in the end, they should conclude their findings with current perspectives and future implications.  5-Authors should be consistent in writing throughout the manuscript and avoid lengthy text that can confuse the reader.  Introduction and discussion should be improved. Again, the topic has its own importance and value. These types of surveys are essential to manage vector-borne diseases. 
